# Neuroprotective Activity of *Mentha* Species on Hydrogen Peroxide-Induced Apoptosis in SH-SY5Y Cells

**DOI:** 10.3390/nu12051366

**Published:** 2020-05-10

**Authors:** Doaa M. Hanafy, Paul D. Prenzler, Geoffrey E. Burrows, Saliya Gurusinghe, Bashar M. Thejer, Hassan K. Obied, Rodney A. Hill

**Affiliations:** 1School of Biomedical Sciences, Charles Sturt University, Locked Bag 588, Wagga Wagga, NSW 2678, Australia; dhanafy@csu.edu.au (D.M.H.); sgurusinghe@csu.edu.au (S.G.); bthejer@csu.edu.au (B.M.T.); hassanpharmacy@yahoo.com (H.K.O.); 2Graham Centre for Agricultural Innovation (an alliance between Charles Sturt University and NSW Department of Primary Industries), Pugsley Place, Wagga Wagga, NSW 2795, Australia; pprenzler@csu.edu.au; 3Department of Pharmacognosy, National Research Centre, Dokki, Cairo 12622, Egypt; 4School of Agricultural & Wine Sciences, Charles Sturt University, Locked Bag 588, Wagga Wagga, NSW 2678, Australia; gburrows@csu.edu.au; 5Department of Biology, Faculty of Science, Wasit University, Al Kut, Wasit 52001, Iraq

**Keywords:** Alzheimer’s disease, amyloid β, β-secretase, Bax, caspase, Lamiaceae, mint, oxidative stress

## Abstract

Alzheimer’s disease (AD) is a progressive neurodegenerative disorder with an unclear cause. It appears that multiple factors participate in the process of neuronal damage including oxidative stress and accumulation of the protein amyloid β (Aβ) in the brain. The search for a treatment for this disorder is essential as current medications are limited to alleviating symptoms and palliative effects. The aim of this study is to investigate the effects of mint extracts on selected mechanisms implicated in the development of AD. To enable a thorough investigation of mechanisms, including effects on β-secretase (the enzyme that leads to the formation of Aβ), on Aβ aggregation, and on oxidative stress and apoptosis pathways, a neuronal cell model, SH-SY5Y cells, was selected. Six *Mentha* taxa were investigated for their in vitro β-secretase (BACE) and Aβ-aggregation inhibition activities. Moreover, their neuroprotective effects on H_2_O_2_-induced oxidative stress and apoptosis in SH-SY5Y cells were evaluated through caspase activity. Real-time PCR and Western blot analysis were carried out for the two most promising extracts to determine their effects on signalling pathways in SH-SY5Y cells. All mint extracts had strong BACE inhibition activity. *M. requienii* extracts showed excellent inhibition of Aβ-aggregation, while other extracts showed moderate inhibition. *M. diemenica* and *M. requienii* extracts lowered caspase activity. Exposure of SH-SY5Y cells to *M. diemenica* extracts resulted in a decrease in the expression of pro-apoptotic protein, Bax, and an elevation in the anti-apoptotic protein, Bcl-xL, potentially mediated by down-regulation of the ASK1-JNK pathway. These results indicate that mint extracts could prevent the formation of Aβ and also could prevent their aggregation if they had already formed. *M. diemenica* and *M. requienii* extracts have potential to suppress apoptosis at the cellular level. Hence, mint extracts could provide a source of efficacious compounds for a therapeutic approach for AD.

## 1. Introduction

Alzheimer’s disease (AD) is a progressive neurodegenerative disorder that is characterized by memory impairment and cognitive decline due to loss of neurons [1]. The pathophysiology of the disease is complicated and not well understood [2]. Accumulation of amyloid β (Aβ) protein outside the neurons in the brain is believed to be the initial event of the disease [3]. Histologically, AD brain is characterized by Aβ plaques and intraneuronal neurofibrillary tangles of hyperphosphorylated-tau [4]. This accumulation disrupts neuron–neuron communication and leads to neuronal death [1]. Aβ formation occurs through abnormal cleavage of amyloid precursor protein (APP) by secretases. Therefore, secretase inhibitors were believed to delay the progression of AD through a reduction in the rate of Aβ generation [5]. Unfortunately, these compounds have not shown efficacy in clinical trials [3]. AD pathology is also associated with oxidative stress [6]. Excess formation of reactive oxygen species (ROS) has been observed in human brains affected by AD as well as in transgenic mouse models of AD [7]. Aβ induces oxidative stress through increasing the formation of ROS [6]. Hence, it has been suggested that antioxidants could be beneficial in AD [7].

Oxidative stress, ROS, and Aβ lead to neuronal death through apoptosis in AD [8]. These apoptotic stimuli activate pro-apoptotic members of the Bcl-2 protein family (Bax (Bcl-2-associated X protein) and Bak (Bcl-2 homologous antagonist killer)) [9]. The apoptotic pathway involves the recruitment and activation of caspases, a family of cysteine-aspartic proteases that play the major role in apoptosis [10]. The apoptotic signalling cascade is regulated by the anti-apoptotic members, Bcl-2 (B-cell lymphoma 2) and Bcl-xL (B-cell lymphoma-extra large), which antagonize the effects of the pro-apoptotic proteins [9]. The Bcl-2-associated death promoter (Bad) protein is another protein that can promote apoptosis indirectly [11]. Active Bad heterodimerizes with anti-apoptotic proteins and blocks their action; therefore, Bad is considered a pro-apoptotic protein. Bad becomes inactive upon phosphorylation by protein kinase B (Akt) [11].

The pathogenesis of AD also includes the activation of apoptosis signal-regulating kinase 1 (ASK1) by ROS and Aβ [8]. ASK1 is a member of the mitogen activated protein kinase kinase kinase (MAPKKK) which activates extracellular signal-regulated kinases (Erk), c-Jun NH_2_-terminal kinases (JNK) and p38 MAPKs pathways [8]. The activation of JNK and p38 MAPK stimulates β-secretase (BACE) causing Aβ accumulation and neuronal cell death [12,13].

The production of ROS is normally balanced by the naturally occurring antioxidant enzymes, such as thioredoxin (Trx), peroxiredoxin-I (Prx) and heme oxygenase-1 (HO-1). The reduced form of Trx can bind to ASK1, which inhibits its activity and prevents apoptosis. Oxidative stress induces the oxidized form of Trx, which cannot bind to ASK1 leading to its activation [8].

There is an urgent need for a prophylactic drug or cure for AD, especially with an increasing number of patients diagnosed with this disease [1]. Plant extracts and their derivatives provide potential therapeutics for AD, particularly those rich in antioxidants and compounds with other bioactivities. Dietary phenolic compounds have been reported to possess antioxidant and neuroprotective properties [6]. Plants from the Lamiaceae have been used traditionally to improve memory [14,15]. *Mentha* (mint), a genus in the Lamiaceae, has been reported to have strong antioxidant and enzymatic inhibition activities relevant to AD and is rich in phenolic compounds [16]. For example, it has been shown that *M.* × *piperita* protected mice from stress, amnesia and neurodegeneration in Aβ-induced models [17].

Here, we present evidence for the potential neuroprotective effect of extracts from different *Mentha* species in vitro. The impact of *Mentha* taxa on the prevention of Aβ formation through BACE inhibition has been evaluated for the first time, as well as on the inhibition of Aβ aggregation. The mechanism by which *Mentha* extracts protected SH-SY5Y cells from H_2_O_2_-induced oxidative damage and apoptosis was examined through their effect on the signalling pathways associated with antioxidant proteins and apoptosis. This study is the first to report the neuroprotective effect of *Mentha* extracts and their possible underlying mechanism of action at the cellular level.

## 2. Materials and Methods

### 2.1. Collection of Plant Material and Preparation of Crude Extracts

Six *Mentha* taxa were purchased from two nurseries in Australia. *Mentha australis* R.Br. (Australian native mint (Ma1-3)), *M. diemenica* Sprengel (Australian native slender mint (Md1-3)), *M. spicata* L. var. *crispa* (Schrad.) Schinz and Thellung (Moroccan mint (MM1-3)), *M.* × *piperita* L. (peppermint (MP1-3)) and *M.* × *piperita* L. var. *officinalis* Sole (white peppermint (WP1-3)) were purchased from Mudbrick Cottage Herb Farm, Mudgeeraba, QLD and *M. requienii* Bentham (Corsican mint (Mr4-6)) was purchased from Greenpatch Organic Seeds and Plants, Glenthorne, NSW. Samples were collected, dried and extracted with aqueous methanol (50% *v*/*v*) under the same conditions so that all results are directly comparable as described earlier [16].

### 2.2. Chemicals and Reagents

HPLC-grade methanol (Burdick & Jackson, Morris Plains, NJ, USA); NaCl (MERCK Pty., Kilsyth, VIC, Australia); 2-amino-2-hydroxymethyl-propane-1,3-diol (Tris), human neuroblast cell line (SH-SY5Y), minimum essential medium eagle (MEM), nutrient mixture F-12 ham, foetal bovine serum (FBS), MEM non-essential amino acid solution (NEAA), L-glutamine solution 200 mM, penicillin-streptomycin solution (Pen-Strep) with 10,000 units penicillin and 10 mg/mL streptomycin, trypsin-EDTA solution, trypan blue, dimethyl sulfoxide (DMSO), 3-(4,5-dimethylthiazol-2-yl)-2,5-diphenyltetrazolium bromide (MTT), Dulbecco’s phosphate buffered saline (PBS), hydrogen peroxide (H_2_O_2_), rosmarinic acid, ascorbic acid, RIPA buffer, Laemmli buffer and Tween 20 were purchased from Sigma-Aldrich (Castle Hill, NSW, Australia). Water used was purified by Barnstead™ GenPure™ xCAD Plus Ultrapure Water Purification System (Thermo Scientific, Waltham, MA, USA).

PureZol RNA isolation reagent, iTaq universal SYBR green supermix, 4–20% Mini-PROTEAN^®^ Precast gels and Immun-blot PVDF (polyvinylidene difluoride) membranes were purchased from Bio-Rad (Gladesville, NSW, Australia). High capacity cDNA reverse transcription kit, Pierce™ protease and phosphatase inhibitor mini tablets and Pierce™ BCA protein assay kit were purchased from Thermo Scientific (Thornton, NSW, Australia). 

SensoLyte 520 β-secretase (BACE) assay kit (fluorimetric) and SensoLyte Thioflavin T amyloid β (1-42) aggregation kit (fluorimetric) were purchased from AnaSpec, USA. BACE and Aβ-aggregation inhibition assays were carried out in 96-well plates, black F96 with lid, flat bottom wells, individually wrapped, sterile (Nunclon delta, Thermo Scientific Nunc, Denmark).

Corning^®^ Costar^®^ 3599, 96- and 24-well cell culture plates, flat bottom (Corning Incorporated, Corning, NY, USA) were used for cell culture.

Caspase-Glo^®^ 3/7 kit was purchased from Promega Corporation, Sydney, NSW, Australia. The assay was carried out using white polystyrene Corning^®^ Costar^®^ 3917, flat bottom with lid, tissue culture treated 96-well plates (Corning^®^ Incorporated, Corning, NY, USA).

### 2.3. Primers and Antibodies

Primers were purchased from Sigma-Aldrich, Castle Hill, NSW, Australia, with the sequences presented in Table 1. Primary polyclonal antibodies, raised in rabbits, were β-actin, Bax, Bcl-2, Bcl-xL, ASK1, p38 MAPK, SAPK/JNK, Erk1/2 and Phospho-Erk1/2 (Thr202/Tyr204) (Cell Signaling Technology, Australian Biosearch, Karinyup, WA, Australia). Anti-rabbit IgG, horseradish peroxidase (HRP)-linked secondary antibody was also purchased from Cell Signaling Technology.

### 2.4. β-Secretase (BACE) Inhibition Activity

The BACE inhibition assay was performed according to the manufacturer’s protocol (screening β-secretase inhibitors using purified enzyme) using SensoLyte 520 β-secretase (BACE) assay kit (fluorimetric) in a cell-free system (AnaSpec, Fremont, CA, USA). The assay was carried out in 96-well plates, black F96 with lid, flat bottom wells, individually wrapped, sterile (Nunclon delta, Thermo Scientific Nunc, Denmark). Briefly, 40 µL human β-secretase (2.5 µg/mL) was added to 10 µL of each extract, followed by 50 µL β-secretase substrate. Crude mint extracts were used at final concentrations in the assay of 50 and 500 µg dry original material/mL solvent. After incubation for 30 min, fluorescence was measured by a fluorescence plate reader (Fluostar Omega BMG Labtech, Germany, software version 1.20) with excitation and emission wavelengths of 490 nm and 520 nm, respectively. LY2886721 (0.25 µM (0.1 µg/mL) final concentration) was used as a reference inhibitor for the enzyme. The assay was performed in triplicate (*n* = 3). Inhibitor activity was expressed as inhibition of relative fluorescence units (RFU) using the formula:(1)$$\%\;\mathrm{inhibition}=\frac{\left({\mathrm{RFU}}_{\mathrm{control}}-{\mathrm{RFU}}_{\mathrm{sample}}\right)\times 100}{{\mathrm{RFU}}_{\mathrm{control}}}$$

### 2.5. Aβ42-Aggregation Inhibition Activity

The amyloid β (1-42) aggregation inhibition assay was performed according to the manufacturer’s protocol using SensoLyte Thioflavin T amyloid β (1-42) aggregation kit (fluorimetric) in a cell-free system (AnaSpec, USA). The assay was carried out in black 96-well plates as above. Briefly, 5 µL of each extract was added to 10 µL 2 mM thioflavin T dye followed by 85 µL Aβ42 peptide solution. Crude mint extracts were used at final concentrations in the assay of 80, 320 and 1280 µg dry original material/mL solvent. The fluorescence intensity was measured by a fluorescence plate reader as above (Section 2.4) with excitation = 460 nm and emission = 485 nm. Phenol red (100 µM (35.4 µg/mL) final concentration) was used as a reference inhibitor. The assay was performed in triplicate (*n* = 3). Inhibitor activity was expressed as inhibition of RFU using the formula:(2)$$\%\;\mathrm{inhibition}=\frac{\left({\mathrm{RFU}}_{\mathrm{control}}-{\mathrm{RFU}}_{\mathrm{sample}}\right)\times 100}{{\mathrm{RFU}}_{\mathrm{control}}}$$

### 2.6. Cell Culture

Human neuroblastoma SH-SY5Y cells, purchased from Sigma-Aldrich passage number 17, were grown in T-75 flasks and incubated at 37 °C/5% CO_2_ in a humidified incubator in complete media composed of 50% MEM, 50% F-12 Ham, 10% FBS, 1% Pen-Strep, 1% L-glutamine and 1% NEAA. When cells reached 80–90% confluency, they were detached from the flask with trypsin-EDTA solution, and subsequently cultured in a fresh medium. Viable and dead cells were quantified using a haemocytometer following the addition of 10% trypan blue solution.

### 2.7. Cell Viability

SH-SY5Y cells were seeded at a density of 5 × 10^4^ cells per well in clear 96-well cell culture plates and maintained at 37 °C/5% CO_2_ in a humidified incubator for 24 h. H_2_O_2_ was freshly prepared prior to each experiment from a 30% stock solution in different concentrations ranging from 100–800 µM. In order to induce oxidative stress, cells were treated with the different concentrations of H_2_O_2_ for 90 min. The cell viability was determined using the conventional colourimetric MTT reduction assay, which is based on the conversion of MTT to purple formazan crystals by mitochondrial dehydrogenase. Briefly, after 90 min exposure to H_2_O_2_, 10 μL of MTT (5 mg/mL in PBS) were added to each well, and the cells were incubated at 37 °C for 60 min [19]. The supernatants were aspirated carefully, and 60 μL of DMSO were added to each well to dissolve the precipitate. The absorbance at 570 nm was measured with a SpectraMax 190 Microplate Reader (Molecular Devices, San Jose, CA, USA) and the data obtained were presented as the percentage of control. The LD_50_ (50% lethal concentration) of H_2_O_2_ toxicity against SH-SY5Y cells was determined.

Moreover, microscopic examination has been used to assess cell viability after treating the cells for 24 h with different concentrations of ascorbic acid, rosmarinic acid or mint extracts. 

### 2.8. Caspase Assay

Detection of caspase-3 and -7 activity was performed according to the manufacturer’s instructions (detection of caspase-3 and -7 activities in cell-based assays) using 96-well cell culture plates. When cells reached 80–90% confluency, they were treated with various concentrations (80, 160, 320, 640 and 1280 μg/mL) of mint extracts for 24 h, prior to exposure to 500 μM H_2_O_2_ for 90 min to detect their neuroprotective activity. The extracts were aspirated before the addition of H_2_O_2_ to prevent direct interaction between mint extracts and H_2_O_2._ To detect the potential use of mint extracts as a treatment, cells were treated with the extracts after exposure to 500 μM H_2_O_2_ for 90 min. Another group of cells was treated, in the same manner, with ascorbic acid as a reference antioxidant compound or rosmarinic acid as the major antioxidant compound present in mint extracts [16]. The control cells were treated with the same medium without H_2_O_2_ or extracts. Luminescence was then detected using a luminometer (Fluostar Omega BMG Labtech, Germany, software version 1.20). The assay was performed in triplicate (*n* = 3). Results have been presented as relative luminescence units (RLU), with luminescence directly proportional to caspase-3/7 activity.

### 2.9. Reverse Transcriptase-Polymerase Chain Reaction (RT-PCR)

When cells reached 80–90% confluency, they were treated with various concentrations of ascorbic and rosmarinic acids (10, 20 and 80 μg/mL) and mint extracts (80, 320 and 1280 μg dry original material/mL solvent) for 24 h. Treatments were then removed, and the cells were exposed to 250 μM H_2_O_2_ for 90 min. The control cells were treated with the same medium without H_2_O_2_ or extracts. Total RNA was extracted from cells cultured in the 24-well plates using PureZol reagent as described by the manufacturer. First-strand cDNAs were synthesised by reverse transcription of 1 µg RNA from each sample. 2 µL of the resulting cDNA was used for real-time PCR with primer sets as mentioned above (Section 2.3) using the conditions: initial denaturation at 95 °C for 3 min, denaturation at 95 °C for 30 s then annealing at 55 °C for 30 s for 39 cycles in a BioRad C1000 thermocycler with a CFX96 real-time fluorescence detection system. Acquired real-time data for each gene target was obtained through Bio-Rad CFX Manager version 3.0. Expression was normalised to that of EF1α expression and was calculated using ΔΔ*C*_t_ method [21] of three independent experiments (*n* = 9).

### 2.10. Western Blot Analysis

Cells were treated with various concentrations of ascorbic or rosmarinic acids (10, 20 and 80 μg/mL) or mint extracts (80, 320 and 1280 μg dry original material/mL solvent) for 24 h in 24-well plates after reaching 80–90% confluency. Treatments were then removed, and the cells were exposed to 250 μM H_2_O_2_ for 90 min. The control cells were treated with the same medium without H_2_O_2_ or extracts. Cells were lysed with RIPA buffer containing protease and phosphatase inhibitors. The lysate was incubated on ice for 5 min and centrifuged at 8000× *g* for 10 min at 4 °C. The supernatant was collected, and the protein concentration was determined using BCA protein assay. Protein lysate was mixed with Laemmli buffer (1:1) and heated at 95 °C for 5 min. Cell lysates were electrophoresed in 4–20% SDS-polyacrylamide gels and transferred onto PVDF membranes. The membranes were incubated overnight at 4 °C with different primary antibodies, which were diluted 1:1000 in TBS-T (0.1% Tween-20 in 1× Tris-buffered saline) containing 5% non-fat milk. Bound antibodies were distinguished by HRP-conjugated anti-rabbit IgG and the band intensities were determined using Bio-Rad Molecular Imager Gel Doc XR+ controlled by Image Lab software version 4.1. Band intensity was analysed by Image Studio Lite software Ver 5.2 from three independent experiments (*n* = 3).

### 2.11. Data Analysis

Data analyses were performed using Microsoft Excel, SigmaPlot 10.0 and R × 64 3.2.3 software. Statistical comparisons were made using analysis of variance (ANOVA) methods and Tukey’s Multiple Comparison Test. The samples were derived from equal variances. All other model assumptions were checked and met (i.e., the residuals had a mean of zero, were normally distributed, had constant variance and were independent). All assays were done in triplicate. A one-way ANOVA model was applied. Data were expressed as the predicted means of each assay ± standard error (SE). Differences between means were considered to be statistically significant at *p* < 0.05. 

## 3. Results

### 3.1. β-Secretase (BACE) Inhibition Activity

Mint extracts showed high BACE inhibition activity, with more than 64% inhibition at 500 µg dry original material/mL solvent final concentration in the assay (Figure 1). MM1-3 and Ma1-3 showed the highest inhibition (93% and 90%, respectively), followed by Md1-3 and WP1-3 (85% and 81%, respectively). At 50 µg dry original material/mL solvent, the extracts had a poor BACE inhibition activity except MM1-3, which exhibited 63% inhibition. The standard inhibitor, LY2886721, had 100% inhibition activity at 0.25 µM (0.1 µg/mL) final concentration.

### 3.2. Aβ42-Aggregation Inhibition Activity

Mint extracts inhibited Aβ42-aggregation at final concentrations of 80, 320 and 1280 µg dry original material/mL solvent (Figure 2). The highest Aβ42-aggregation inhibition was exhibited at 1280 µg/mL of Mr4-6 (93%). The standard inhibitor, phenol red, inhibited Aβ42-aggregation by 52% at 100 µM (35.4 µg/mL) final concentration. The three concentrations of MM1-3 and WP1-3, 320 µg/mL Mr4-6 and 320 and 1280 µg/mL WP1-3 had inhibition activities that ranged from 29% to 53%.

### 3.3. Cell Viability

SH-SY5Y cells were exposed to H_2_O_2_ (100–800 µM) for 90 min and cell survival was assessed by MTT assay. Significant decreases in cell survival were induced by 100 to 800 µM H_2_O_2_ in a dose-dependent manner (Appendix A) with an LD_50_ of approximately 500 µM (*n* = 3). Therefore, caspase assay was performed with the treatment of 500 µM H_2_O_2_ for 90 min. This concentration is higher than the physiological level of H_2_O_2_, but it is within the range previously used (100–1000 µM) [22,23] as a step for studying potential neuroprotective agents; therefore, it was used to induce SH-SY5Y cell injury. To ensure that cell injury was caused only by H_2_O_2_ not by other treatments in subsequent assays, microscopic examination (Appendix A) showed that treating the cells for 24 h with up to 1280 µg/mL mint extracts or ascorbic acid had no negative effect on cell viability, as well as up to 320 µg/mL rosmarinic acid. Hence, these concentrations were the maximum used in the assays.

### 3.4. Effects of Mint Extracts on Caspase Activity

The effectiveness of mint extracts as neuroprotective and their potential use as a treatment were further evaluated by determining the mechanism of apoptosis through caspase activation. The Caspase-Glo 3/7 assay is a luminescent assay that measures caspase-3 and -7 activities. To evaluate the neuroprotective potential of mint extracts (Figure 3), SH-SY5Y cells were pretreated with five concentrations (80, 160, 320, 640 or 1280 µg/mL) of mint extracts for 24 h. Upon completion of the treatment period, cell damage was induced using 500 μM of freshly prepared H_2_O_2_. As shown in Figure 3, 500 μM H_2_O_2_ increased the activity of caspases-3/7. Cells pretreated with Md1-3 showed a decrease in caspases-3/7 activity in a dose-dependent manner (*p* < 0.05) indicating a potential neuroprotective activity. Pretreatment of the cells with the three highest concentrations (320, 640 or 1280 µg/mL) of Mr4-6 resulted in a significant (*p* < 0.05) decrease in caspases-3/7 activity. Rosmarinic acid at 320 µg/mL (890 µM) and all concentrations of ascorbic acid used were also able to protect the cells from H_2_O_2_-induced damage and significantly decreased the activity of caspases-3/7. The potential use of mint extracts as a treatment was also evaluated by treating the cells with mint extracts after exposure to H_2_O_2_ for 90 min with none of the extracts being able to reverse the damage (Appendix A).

### 3.5. Effect of Mint Extracts on Transcriptomic Regulation of Apoptotic and Antioxidant Genes in SH-SY5Y Cells Exposed to H_2_O_2_

As Md1-3 and Mr4-6 showed potential neuroprotective effect by reducing caspase activity, quantitative real-time PCR was performed on total RNA isolated from pretreated cultured cells to quantify apoptotic and antioxidant gene expression levels. Cells were cultured in 24-well plates in which cells were more sensitive to 500 µM H_2_O_2_ than in 96-well plates, therefore, cell damage was induced using 250 µM H_2_O_2_. H_2_O_2_-treatment induced upregulation of the pro-apoptotic gene, Bax, in SH-SY5Y cells in comparison to untreated cells (Figure 4). Pretreatment of SH-SY5Y cells with the highest concentration of ascorbic acid (80 μg/mL = 450 μM) or rosmarinic acid (80 μg/mL = 220 μM) prior to H_2_O_2_ exposure increased the expression of the anti-apoptotic genes Bcl-2 and Bcl-xL more than with H_2_O_2_ exposure alone or in untreated cells (*p* < 0.05), while only 80 μg/mL of ascorbic acid increased the expression of the antioxidant genes Trx and HO-1.

Pretreating the cells with the mint extract Md1-3 (80, 320 or 1280 μg/mL) resulted in downregulation of the expression of Bax (*p* < 0.05). Pretreatment with 1280 μg/mL Md1-3 resulted in upregulation of the expression of the anti-apoptotic gene Bcl-xL and the antioxidant gene Prx, more than with H_2_O_2_ exposure alone or in untreated cells (*p* < 0.05). Cells pretreated with 80 or 320 μg/mL Mr4-6 showed a significant decrease in the expression of the pro-apoptotic gene, Bad (*p* < 0.05), while the concentration 1280 μg/mL showed no change in pro-apoptotic gene expression.

### 3.6. Effect of Mint Extracts on the Protein Expression in SH-SY5Y Cells Exposed to H_2_O_2_

Western blot analysis was used to further investigate the effect of Md1-3 and Mr4-6 extracts, rosmarinic acid and ascorbic acid on protein expression and signalling pathways of apoptosis. Their effects on the expression of the proteins Bax and Bcl-xL was similar to those induced on mRNA expression (Figure 5 and Appendix A). Cell damage induced by H_2_O_2_ increased the expression levels of both ASK1 and Erk1/2 significantly, while there was apparently a numeric but non-significant increase in expression of p38 MAPK, SAPK/JNK and P-Erk1/2 (Figure 5 and Figure 6 and Appendix A). Md1-3 at 320 and 1280 μg/mL significantly decreased the expression level of ASK1, while only 320 and 1280 μg/mL Mr4-6 significantly decreased the expression level of SAPK/JNK and Erk1/2, respectively. However, an apparent numeric but non-significant decrease in the expression levels of p38 MAPK, Erk1/2 and P-Erk1/2 was observed upon pretreating the cells with the other concentrations of Md1-3 or Mr4-6.

## 4. Discussion

In a previous study [16], 19 *Mentha* taxa were screened for in vitro bioactivities linked to AD which included: in vitro antioxidant capacity, acetylcholinesterase (AChE), butyrylcholinesterase (BuChE) and histone deacetylase inhibition. To cover the broad spectrum of bioactivities exhibited by these mints and on the basis of principal component analysis (PCA) (see Figure 1 in [16]), six *Mentha* taxa were selected for further investigation. MM1-3 and Mr4-6 had the highest phenolic content, antioxidant capacity and HDAC inhibition; WP1-3 had the highest AChE inhibition; and Ma1-3, Md1-3 and MP1-3 had more moderate activities. In addition, MP1-3 (peppermint) is the taxon that is most frequently consumed in food and drinks [16].

Extracellular deposition of amyloid (senile) plaques is considered the main pathological feature of AD [1], although it is not exactly clear whether Aβ accumulation is the cause of AD or if it is produced as a consequence of neuronal damage of unknown cause [3]. Amyloid plaques are aggregates of Aβ40 and Aβ42 formed through the proteolytic cleavage of APP by secretases (β (BACE) and γ) [9]. BACE knockout mice showed a diminished production of Aβ [24]. Therefore, inhibition of BACE and/or inhibition of Aβ-aggregation could provide a potential therapy for AD. LY2886721 is a potent BACE inhibitor that reduced BACE activity by 50–75% and Aβ42 by 72% in phase I clinical trials but phase II trials were terminated because of liver toxicity in some patients [25]. Although most of the BACE inhibitors, such as verubecestat (MK-8931), were efficient in suppressing Aβ production in preclinical and clinical trials, they showed failure in phase III in mild to moderate AD patients due to toxicity or cognitive worsening compared to placebo-treated patients [4,26]. However, clinical trials for the BACE inhibitor E2609 (elenbecestat) are still active and expected to be completed by 2023 [2] (https://clinicaltrials.gov/ (accessed on 30 April 2020)). It is also suggested that BACE inhibitors could be potentially effective when combined with other therapeutic agents that target different pathogenic mechanisms [4].

The inhibitory effect of mint extracts on BACE and Aβ42-aggregation had not been previously studied. The six mint extracts under investigation showed excellent BACE and Aβ42-aggregation inhibition at 500 µg/mL (64% for Mr4-6; 93% for MM1-3) and at 1280 µg/mL (22% for MP1-3; 93% for Mr4-6), respectively. While the concentrations used in the present study are relatively high, we hypothesize that this BACE inhibition activity is one of a complex of multiple effects. The possibility of identifying novel BACE inhibitors is quite intriguing. Mint extracts were rich in biophenols [16], and it has been found that natural flavonoids had BACE inhibitory activity in vitro [27]. Salvianolic acid, a phenolic acid present in mints, showed an effect against BACE [28]. Furthermore, several phenolic compounds were found to inhibit Aβ-aggregation in vitro [29] and improve performance on learning and spatial memory tasks in a transgenic mice model of AD [28,30,31,32]. Rosmarinic acid, salvianolic acid and an extract from *Salvia officinalis* (Lamiaceae) reduced cell death induced by Aβ42 [28,33,34]. Freeze-dried extract of *M. piperita* protected Aβ- and ageing-induced mice from stress, amnesia and neurodegeneration and improved acquisition and retention in behavioural models [17].

However, the activities of mint extracts cannot be assigned to a single class of compounds. Ma1-3 and Md1-3 possessed moderate total phenolic contents (TPC) (5.5 and 5.7 mg gallic acid equivalent (GAE)/g DW (dry weight)) [16], but they showed a strong BACE inhibitory activity (91% and 86%), respectively. Mr4-6 had higher TPC (8.6 mg GAE/g DW) and a lower BACE inhibitory activity (68%) but a higher Aβ42-aggregation inhibition activity (93%). The activities of mint extracts are probably due to a combination of several classes of compounds that present together in the extract rather than a single compound or a single class.

AD is characterized by brain shrinkage due to programmed cell death/apoptosis [9]. Therefore, stimulation of endogenous anti-apoptotic pathways might provide a potential therapy by limiting apoptosis in neurodegenerative diseases such as AD [10]. This can be achieved by suppressing pro-apoptotic proteins and enhancing anti-apoptotic proteins.

H_2_O_2_, as a major generator of ROS, can induce apoptosis and cell death in various cell lines [35]. As apoptotic cell death is associated with the activation of caspases, in the present study, the activity of caspases was first assessed as an indication of apoptosis. The human neuroblastoma SH-SY5Y cell line is a widely used in vitro model to study neuronal function and oxidative stress-induced neuronal cell death [36,37,38]. SH-SY5Y cells treated with H_2_O_2_ showed an increase in caspase activity. Ascorbic acid and rosmarinic acid were used as positive controls being known antioxidant compounds. Pretreating the cells with ascorbic acid showed a massive decrease in caspase activation, which is consistent with the findings of Huang and May [39]. Only the highest tested concentration of rosmarinic acid (320 µg/mL, 890 µM) decreased the activation of caspases under our experimental conditions. It has been found in another study that 56 µM rosmarinic acid could decrease the activation of caspases in H_2_O_2_-induced apoptosis in SH-SY5Y cells [35]. Md1-3 and Mr4-6 extracts were able to decrease the activation of caspases-3/7, which indicates a potential neuroprotective activity; hence, it was deemed of interest to further investigate their mechanism of action and their effect on gene regulation and protein signalling.

SH-SY5Y cells have been used to study the signalling pathways of different compounds such as curcumin and propofol [22,23]. Treating the cells with H_2_O_2_ increased the expression of pro-apoptotic Bax as well as anti-apoptotic Bcl-xL. The rise in the expression levels of both genes may suggest that H_2_O_2_ treatment stimulates the entire pro-apoptotic and anti-apoptotic axes, which may then be regulated through other mechanisms including modulation via other regulatory RNAs or activation or sequestration of pathway molecules through phosphorylation/dephosphorylation of proteins in the regulatory response to protect the cells from apoptosis [40]. Pretreating the cells with Md1-3 resulted in downregulation of the expression level of Bax and upregulation of the expression level of Bcl-xL at both mRNA and protein levels, providing evidence for its role in suppressing apoptosis. H_2_O_2_ as well as 80 µg/mL ascorbic acid increased the expression of HO-1. It was found that oxidative stress and non-toxic phytochemicals and drugs mediate HO-1 expression [41,42]. The upregulation of the expression level of the antioxidant gene Prx by 1280 μg/mL Md1-3 possibly represents induction of an endogenous antioxidant defence mechanism against H_2_O_2_-induced oxidative stress and apoptosis. The phosphorylation status of Bad determines its activity [11], hence, the effect of Mr4-6 on Bad could be due to an effect on its phosphorylation. To elucidate the signal transduction pathways involved in the effect of mint extracts on gene regulation, we examined their effect on upstream kinases.

ROS, including H_2_O_2_, play a dual role in redox regulation of cellular signalling pathways [43]. This means that they can induce cell death or cell survival according to their effect, whether it is upstream or downstream of the transcription factors of the signalling cascades. For example, H_2_O_2_ activates the three members of MAPK family: p38 MAPKs, JNK and Erk1/2. Activation of p38 MAPKs and JNK induces cell death [43] (Figure 7), while activation of Erk1/2 generally promotes cell survival and proliferation [44]. JNK can promote apoptosis by enhancing the expression level of pro-apoptotic genes (Bax) and suppressing the effect of the anti-apoptotic genes (Bcl-2 and Bcl-xL) by phosphorylation [45]. In addition, activation of p38 MAPKs and JNK leads to Tau hyperphosphorylation, one of the major hallmarks in AD-affected brains, and stimulates BACE causing Aβ42 deposition and neuronal cell death in AD [12]. Meanwhile, activation of Erk1/2 can promote cell survival through activation of the anti-apoptotic proteins, which occurs by downstream phosphorylation of the pro-apoptotic protein, Bad [44]. Phosphorylation of Bad sequesters Bad in the cytoplasm and blocks its interaction with Bcl-2 or Bcl-xL making them available to exert their anti-apoptotic effect [44]. It has been reported that the three MAPKs can induce the expression of a variety of antioxidant enzymes such as HO-1 through the phosphorylation of nuclear factor (erythroid-derived 2)-like 2 (Nrf2) [42]. Phosphorylation of Nrf2 promotes its translocation into the nucleus where it mediates HO-1 gene transcription [41]. ASK1 is a member of the MAPK pathway invoking activation of the Erk, JNK and p38 MAPKs pathways (Figure 7). It can be activated by ROS and mediate cell death [8]. Normally, the reduced form of the antioxidant enzyme, Trx1, binds to ASK1 which blocks its kinase activity and inhibits apoptosis. However, ROS oxidize Trx1 leading to the dissociation of the Trx1-ASK1 complex and activation of ASK1, which induces apoptosis [8]. Prx can have both positive and negative effects on the activation of MAPK pathway [46].

In the present study, cell damage induced by H_2_O_2_ increased the expression levels of both ASK1 and Erk1/2 significantly, while it appeared that there was a numerical but non-significant increase in p38 MAPK, SAPK/JNK and P-Erk1/2. It has been previously reported [44] that an increase in the expression levels of SAPK/JNK, and Erk1/2 in SH-SY5Y is induced by H_2_O_2_, while an increase in p38 MAPK has been reported by others [47]. The effect of H_2_O_2_ on transcription factors varies by different concentrations and different times of exposure as well as different cell types [48]. The down-regulation of Bax and the up-regulation of Bcl-xL exerted by pretreating the cells with Md1-3 could be through the significant down-regulation of ASK1, which led to down-regulation of the JNK signalling cascade, but not Erk1/2, although non-significant under our experimental conditions. The down-regulation of ASK1 could be due to the up-regulation of Prx by Md1-3. It has been reported by Latimer and Veal [46] that Prx inhibits the activation of MAPK by removing peroxides. The increase in the expression level of Trx and HO-1 by Md1-3 could be useful as a natural defence mechanism to protect cells against H_2_O_2_-induced insult. This could be explained by its activation to Erk1/2, which phosphorylated Nrf2, which in turn mediated HO-1 gene transcription. The effect of Mr4-6 on Bad could be due to an effect on other kinases rather than the MAPK pathway.

It was reported previously that the six mint extracts under investigation had antioxidant, acetylcholinesterase and histone deacetylase inhibition activities [16]. The present study expands these activities to BACE and Aβ-aggregation inhibition activities. Among these extracts, *M. diemenica* and *M. requienii* lowered caspases-3/7 activity indicating a potential role in suppressing apoptosis at the cellular level. Combining all these activities together we find that *M. diemenica* and *M. requienii* target multiple causes of AD.

## 5. Conclusions

AD is a neurodegenerative disorder that currently has no treatment. The neuroprotective effects of extracts from *Mentha* species at the cellular level have not been investigated previously. All mint extracts in the present study showed a strong BACE inhibition activity indicating their potential to prevent the formation of senile plaques. *M. requienii* extract showed an excellent inhibition of Aβ-aggregation, while other extracts showed a moderate inhibition. *M. diemenica* and *M. requienii* extracts showed a potential role in suppressing apoptosis in H_2_O_2_ stressed SH-SY5Y cells as indicated by a decrease in caspase activity. The mechanism underlying this effect could be through a decrease in the expression level of pro-apoptotic protein, Bax, and/or an elevation in the anti-apoptotic protein, Bcl-xL, mediated by a down-regulation of the ASK1-JNK pathway. Nevertheless, the underlying mechanisms are certainly more complex than described here and require further investigation. This study is the first at the cellular level to address the possibility of mint extracts to be used for the prophylaxis from AD or as the basis of a treatment, and based on our findings, further in vivo and/or clinical trials would appear warranted to further explore the potential of mint extracts for treating AD.

## Figures and Tables

**Figure 1 nutrients-12-01366-f001:**
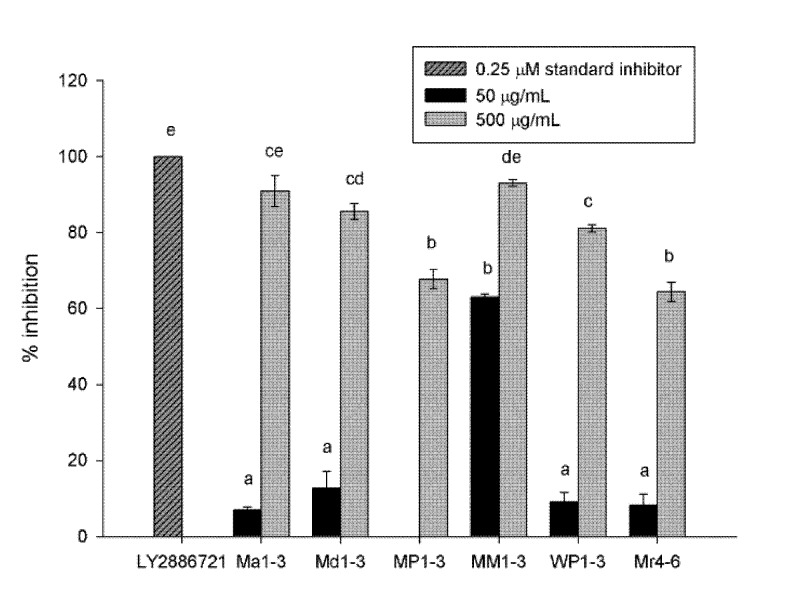
Percentage inhibition of β-secretase (BACE) of *Mentha australis* (Ma1-3), *M. diemenica* (Md1-3), *M. spicata* var. *crispa* (MM1-3), *M.* × *piperita* (MP1-3), *M.* × *piperita* var. *officinalis* (WP1-3) and *M. requienii* (Mr4-6). Results are presented as mean ± standard error (SE) (*n* = 3). Different letters indicate a significant difference at *p* < 0.05. 50 µg/mL MP1-3 resulted in no inhibition.

**Figure 2 nutrients-12-01366-f002:**
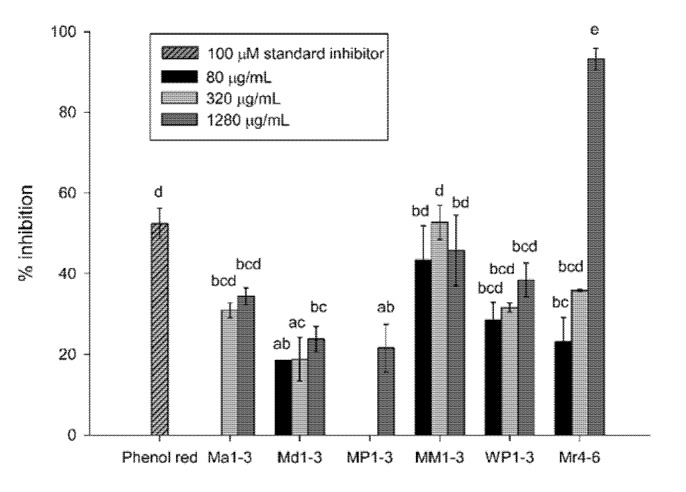
Percentage inhibition of Aβ42 (1-42) aggregation of *Mentha australis* (Ma1-3), *M. diemenica* (Md1-3), *M. spicata* var. *crispa* (MM1-3), *M.* × *piperita* (MP1-3), *M.* × *piperita* var. *officinalis* (WP1-3) and *M. requienii* (Mr4-6). Results are presented as mean ± standard error (SE) (*n* = 3). Different letters indicate a significant difference at *p* < 0.05. 80 µg/mL Ma1-3 and 80 µg/mL and 320 µg/mL MP1-3 resulted in no inhibition.

**Figure 3 nutrients-12-01366-f003:**
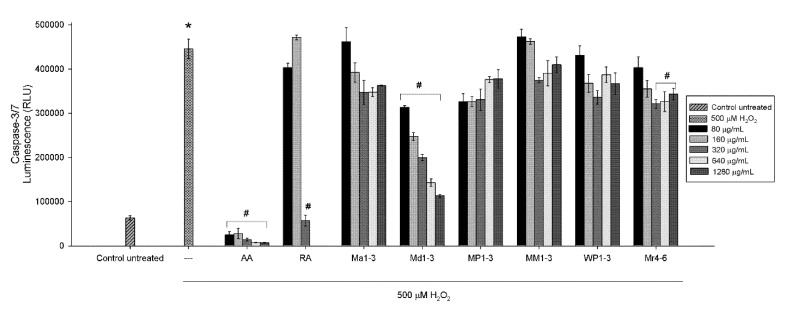
Neuroprotective effect of different concentrations of ascorbic acid, rosmarinic acid and mint extracts on caspases-3/7 activity in H_2_O_2_-induced damage in SH-SY5Y cells. Results are presented as mean ± standard error (SE) (*n* = 3). * indicates that treatment with H_2_O_2_ is significantly different (*p* < 0.05) from control untreated cells, # indicates that pretreatment with mint extracts, AA or RA is significantly different (*p* < 0.05) from H_2_O_2_-treated cells. AA: ascorbic acid, RA: rosmarinic acid, Ma1-3: *Mentha australis*, Md1-3: *M. diemenica*, MP1-3: *M.* × *piperita*, MM1-3: *M. spicata* var. *crispa*, WP1-3: *M.* × *piperita* var. *officinalis* and Mr4-6: *M. requienii*.

**Figure 4 nutrients-12-01366-f004:**
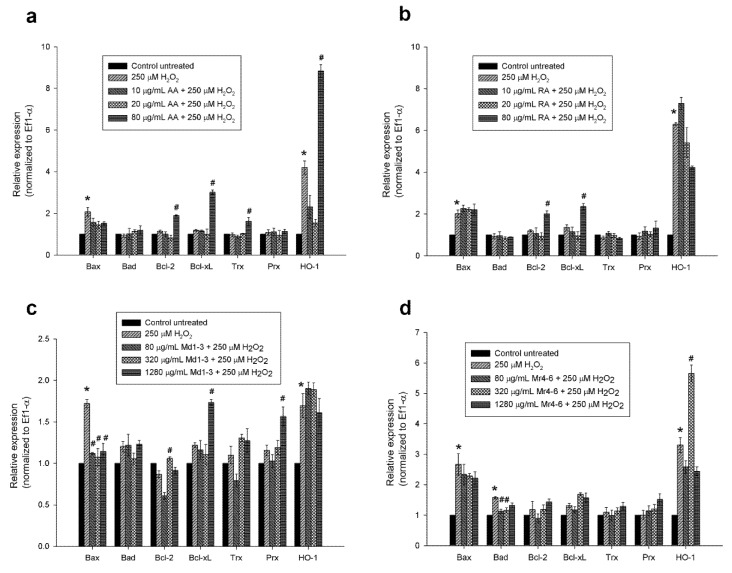
Effect of different concentrations of (**a**) ascorbic acid; (**b**) rosmarinic acid; (**c**) *M. diemenica* and (**d**) *M. requienii* on the mRNA expression of Bax, Bad, Bcl-2, Bcl-xL, thioredoxin, peroxiredoxin-I and heme oxygenase-1 in H_2_O_2_-induced damage in SH-SY5Y cells. Results are presented as mean ± standard error (SE) (*n* = 3). * indicates that treatment with H_2_O_2_ is significantly different (*p* < 0.05) from control untreated cells within a single gene, # indicates that pretreatment with mint extracts, AA or RA is significantly different (*p* < 0.05) from H_2_O_2_-treated cells within a single gene. AA: ascorbic acid, RA: rosmarinic acid, Md1-3: *Mentha diemenica*, Mr4-6: *M. requienii*, Trx: thioredoxin, Prx: peroxiredoxin-I, HO-1: heme oxygenase-1.

**Figure 5 nutrients-12-01366-f005:**
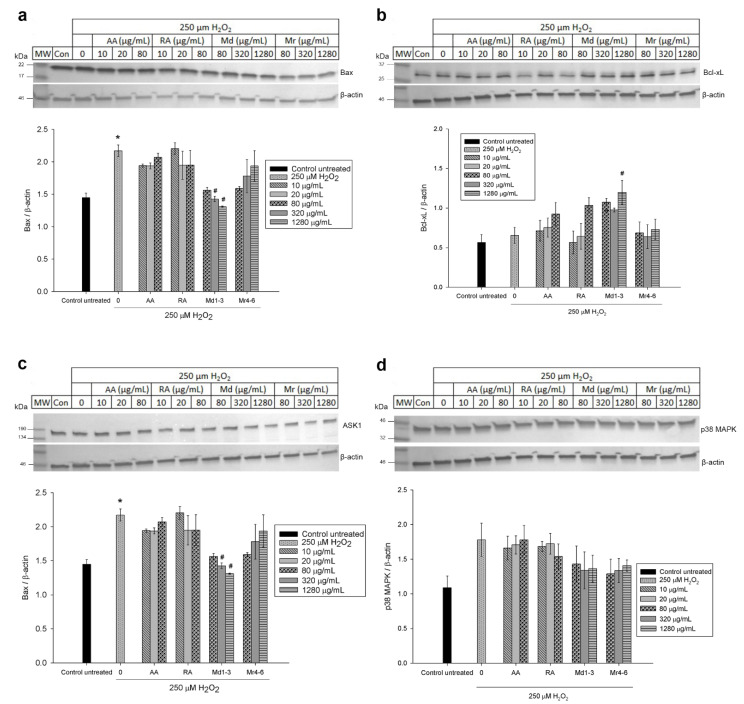
Western blot analysis of different concentrations of ascorbic acid, rosmarinic acid and mint extracts on the expression of (**a**) Bax; (**b**) Bcl-xL; (**c**) ASK1 and (**d**) p38 MAPK in H_2_O_2_-induced damage in SH-SY5Y cells. Results are presented as mean ± standard error (SE) (*n* = 3). * indicates that treatment with H_2_O_2_ is significantly different (*p* < 0.05) from control untreated cells, # indicates that pretreatment with mint extracts, AA or RA is significantly different (*p* < 0.05) from H_2_O_2_-treated cells. AA: ascorbic acid, RA: rosmarinic acid, Md1-3: *Mentha diemenica* and Mr4-6: *M. requienii*.

**Figure 6 nutrients-12-01366-f006:**
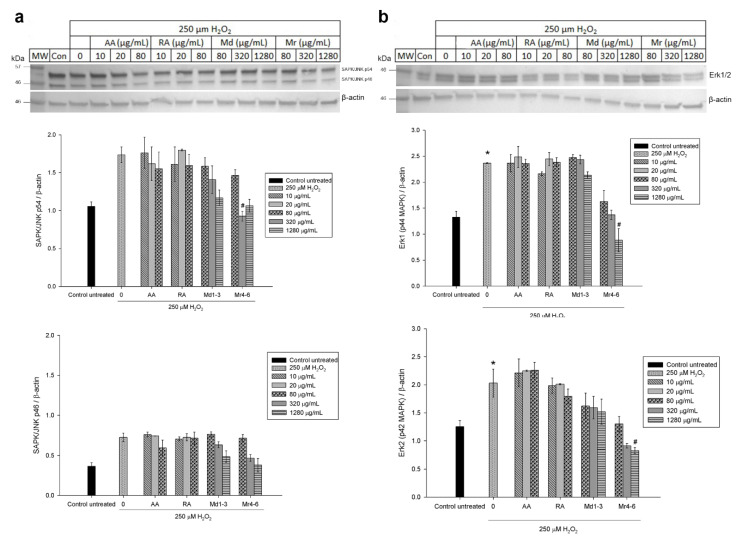
Western blot analysis of different concentrations of ascorbic acid, rosmarinic acid and mint extracts on the expression of (**a**) SAPK/JNK and (**b**) Erk1/2 (p44/42 MAPK) in H_2_O_2_-induced damage in SH-SY5Y cells. Results are presented as mean ± standard error (SE) (*n* = 3). * indicates that treatment with H_2_O_2_ is significantly different (*p* < 0.05) from control untreated cells # indicates that pretreatment with mint extracts, AA or RA is significantly different (*p* < 0.05) from H_2_O_2_-treated cells. AA: ascorbic acid, RA: rosmarinic acid, Md1-3: *Mentha diemenica* and Mr4-6: *M. requienii*.

**Figure 7 nutrients-12-01366-f007:**
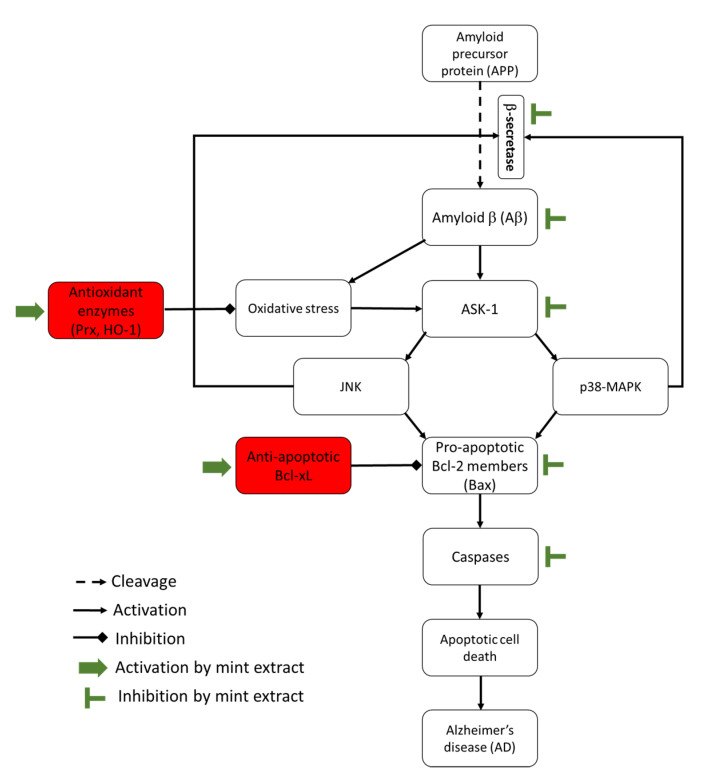
This relatively simplistic diagram summarises the potential effects of mint extracts on the pathways leading to development of Alzheimer’s disease (AD).

**Table 1 nutrients-12-01366-t001:** Forward and reverse primer sequences used in real-time Reverse Transcriptase-Polymerase Chain Reaction (RT-PCR).

Gene Name	Primer Sequences	Reference
Elongation factor 1-alpha (EF1α)	F: 5′-CTGAACCATCCAGGCCAAAT-3′ R: 5′-GCCGTGTGGCAATCCAAT-3′	[18]
Bcl-2-associated × protein (Bax)	F: 5′-GGGGACGAACTGGACAGTAA-3′ R: 5′-CAGTTGAAGTTGCCGTCAGA-3′	[19]
Bcl-2-associated death promoter (Bad)	F: 5′-CCCAGAGTTTGAGCCGAGTG-3′ R: 5′-CCCATCCCTTCGTCGTCCT-3′	[20]
B-cell lymphoma 2 (Bcl-2)	F: 5′-CGACTTCGCCGAGATGTCCAGCCAG-3′ R: 5′-ACTTGTGGCCCAGATAGGCACCCAG-3′	[19]
B-cell lymphoma-extra large (Bcl-xL)	F: 5′-TTCAGTGACCTGACATCCCA-3′ R: 5′-TCCACAAAAGTATCCCAGCC-3′	[19]
Thioredoxin (Trx)	F: 5′-TGCTTTTCAGGAAGCCTTG-3′ R: 5′-TGTTGGCATGCATTTGACTT-3′	[19]
Peroxiredoxin-I (Prx)	F: 5′-TGCCAGATGGTCAGTTTAAA-3′ R: 5′-CAGCTGGGCACACTTCCCCA-3′	[19]
Heme oxygenase-1 (HO-1)	F: 5′-CACGCCTACACCCGCTACCT-3′ R: 5′-TCTGTCACCCTGTGCTTGAC-3′	[19]

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
