# Peer review of "Neuroprotective Activity of Mentha Species on Hydrogen Peroxide-Induced Apoptosis in SH-SY5Y Cells"

_nutrients, 2020, doi:10.3390/nu12051366_

Round 1
Reviewer 1 Report
The manuscript by Hanafi DM et al. provides useful information about the potential use of some mint extracts in the therapeutic approach for Alzheimer's disease. The Authors investigated the mechanism of action underlying the tested mint extracts, providing a deeper knowledge about the bioactivity of these compounds.
After the detailed manuscript revision carried out by the Authors in line with Reviewer's 1 and 2 comments, it seems to me that the paper sounds scientifically robust, well written and clear, and ready to be published without additional intervention.
Author Response
Many thanks to the reviewer. your comments really helped us to improve the manuscript.
much appreciated.
Rod Hill
Reviewer 2 Report
This is a nicely written manuscript and addresses neuroprotective activity of mentha species in a model experiment of H2O2 -induced apoptosis in SH-SY5Y cells that is acceptable for publication in Nutrients with spell check and correction of grammatical errors.
Line 24 - "(the enzyme the leads…" needs correction. "the" should be changed to "that".
Author Response
correction now included in revised manuscript
Reviewer 3 Report
This manuscript describes the in vitro neuroprotective effect of Mentha (mint) extracts in SH-SY5Y cells and the possible underlying mechanisms of action at the cellular level. Authors tested the mint extracts on beta-secretase (BACE1) activity, the enzyme responsible for the first metabolic step that leads to the formation of b-amyloid (Aβ), a 42-amino acid peptide implied in the pathophysiology of Alzheimer’s disease (AD). Mentha is rich in phenolic compounds and has been reported to have antioxidant activity. M. x piperita was shown to protect mice from stress, amnesia and neurodegeneration in Aβ-induced models.
The studies appear to be technically correct but the high mint extract concentrations used limit their pharmacological value for future in vivo studies. Most importantly, Authors appear not aware the all BACE1 inhibitors have been discontinued for the treatment of AD because were not effective and in several cases caused worsening of cognition and behavior of patients, compared to placebo (Egan et al. N Engl J Med 2019; 380: 1408-1420, Henley et al. N Engl J Med 2019; 380: 1483-85). Thus, the claimed inhibitory activity of mint extracts on BACE1 activity does not appear attractive from a clinical point of view.
Specific comments are:
- In vitro, mint extracts showed 64% in BACE inhibitory activity at 500 μg/mL concentration. It is quite unlike to reach such high concentrations in vivo. Authors should discuss this limitation. As reference, the positive control that Authors used (the BACE inhibitor LY2886721) showed 100% inhibition activity at 0.1 μg/mL concentration (0.25 μM).
- Similarly, for Aβ42-aggregation active minta extract concentrations (80, 320 and 1280 μg/mL) were quite high and not easily achievable in vivo.
- The experiments on anti-pro-apoptotic activity of minta extracts are interesting but the use of high concentrations (250-500 μM) of H2O2 to induce apoptotic activity, limit their pharmacological interpretation.
- Authors should mention that all seventeen BACE-inhibitors entering the clinic were found to be ineffective in AD patients and, in most cases worsened, compared to placebo, cognitive and clinical performance of patients (Imbimbo & Watling. Expert Opin Investig Drugs 2019; 28: 967-75).
- In the abstract, I believe that the sentence “Alzheimer’s disease… develops as a consequence of different factors such as oxidative stress and accumulation of the protein amyloid β (Aβ) in the brain” is quite imprudent based on 20-year long multiple failures of clinical trials with different categories of anti-Aβ drugs (Panza et al. Nat Rev Neurol 2019; 15: 73-88). Similarly, in the Introduction, the sentence “One of the main causes of AD is the accumulation of amyloid β (Aβ) protein outside the neurons in the brain” appears quite reckless.
- In the Introduction the sentence “Therefore, secretase inhibitors could delay the progression of AD through a reduction in the rate of Aβ generation” and the cited supporting reference (# 2), are outdated since both gamma-secretase and beta-secretase inhibitors were found ineffective and even detrimental in patients with AD at any clinical stage (Knopman. Nat Rev Neurol 2019; 15: 61-62).
Author Response
No further changes have been recommended to the R1 version.
many thanks
Rod Hill
This manuscript is a resubmission of an earlier submission. The following is a list of the peer review reports and author responses from that submission.
Round 1
Reviewer 1 Report
Abstract
There needs to be a clearly stated hypothesis and, or, aim that is to be tested.
Introduction
What are the Authors suggesting is the source of ROS in AD? Why have the Authors not measured oxidised Trx directly in their studies, rather than the downstream events? It is not clear how they believe the menthol extracts work. They seem to suggest that it is through preventing oxidation of TRx, stopping activation of ASKI and subsequently downstream kinases. This needs to be made clearer. Perhaps a diagram might be useful to explain the theory here? Also, how is the measurement of BACE and Ab- aggregation related to ROS and the cell signalling pathways referred to? Are the authors looking at two unrelated phenomena? Line 81-82: The Authors claim that “This study is the first to report the neuroprotective effect of Mentha extracts and their possible underlying mechanism of action at the cellular level”, however their previous publication (ref 13) as well as work by several other authors have demonstrate anti-oxidant effects of Mentha extracts, including work in cells lines and animals? What seems novel is demonstrating this effect in Neuroblastoma cells. As they have not measured Trx directly they can not be certain of the mechanism either. The Authors need to be more accurate with their claims.
Materials and Methods
The Authors need to consider that previous work of plant extracts has shown that the content can vary from growth conditions and climate, making it unclear how much effect is due to species difference. Also, that the efficiency of extraction can vary, even when the same method is used. What precautions were taken, or tests made, to ensure the extraction of the active, as opposed to total, material was the same between the different species preparations? Line 88; publication 13 is not freely available to readers. and thus, referring readers to a figure in it is pointless. Line 95-96; the comments on results seem a bit out of place in Material and Methods, as do some other statements here? Such comments and explanations may better placed in the introduction. Where were the SH-SY5Y cells sourced from? What passage numbers were used? The Authors should make it clear that this is tumour-cell line and, as such, displays a number of genetic aberrations due to its cancerous origin. The Authors need to reflect on this in their discussion and conclusions. It is not clear if cell were preincubated with the Mentha extracts, which was then removed prior to exposure to H2O2, or whether the extract was left in during the exposure. The Authors should also to provide a rationale for this either way. The Authors need to describe the basis to the BACE and A42 tests used. What actual cell incubations were these assays carried out on? To what extent is the H2O2 is reacting with the external contents of the media, including the Mentha extracts, before the actual cells? Thus they are looking at cells responding to what active H2O2 , or its products, that are left over? The Authors need to make this clearer in their results and discussion. The Authors need to clearly state which size plates were used for which assays. Why were two different sized plates used, i.e. 24 and 96-well, rather than bulking up wells from the smaller plate? What extent of confluency had the cells reached in the well prior to an experiment? Buking up material from individual wells would also have removed the need to use both 250 mM and 500mM H2O2, , which is quite different in toxicity, in different experiments, making it difficult to cross-compare the The Authors need to clearly state how many replicates were used in each experiment, for example how many wells made up an experiment and how many times was each experiment repeated? How many replicates were used for each assay? A fundamental problem with these experiments is the very low number of replicates used, i.e. only three. Moreover, this appears to refer to replicate well in a plate, which are normally used to control experiment variability, rather than using three plates or separate experiments. The low number of replicates, even if they were true replicates, make statistical analysis inappropriate. The use of SEMs, rather than SDs, given that the Authors state that the data is normally distributed, suggests that the variability was quite high too.Results.
It is not clear in Fig.1 what the different letters indicating significance actual mean? Similarly, in the other figures. It is not clear what hypothesis was statically tested in each figure. It looks like everything is being tested against everything else which, if true, is not appropriate. In figure 3; according to the methods, to quote “The cells were treated with…mint extracts for 24 h, prior to or after exposure to 500 μM H2O2 for 90 min”. This means that there should be statistical results for each experiment, the ones for cells treated for 24 hours prior to exposure and the ones treated after exposure, However, only one set appear to have been presented, or is the initial description wrong? Figure 4 is very small and thus difficult to read. I do not understand the “Y” axis values or ho they have obtained? It says they are normalized to Ef-1a? . Where is this described Figure 5d seems to show differences, similar to the other graphs , but has no letters on the bars? It is not clear (line 244) why the MTT assay has been used until this point, when microscopic examination (less reliable) is now used instead? The method for doing this has not been described either. The actual picture provides could be interpreted to actually show a reduction in cells numbers with exposure to the extracts. Line 248; methods seem to be confused with result here? It would be better to have all the methods together under the methods section. It makes it much easier to concentrate on the results that way. It also not always clear what control means in each of the graphs, as this has not been clearly been stated in the methods.Discussion.
The discussion is far too long and tends to become more of a speculative review than a strict interpretation of the meaning of the results. Similarly, 45 references seem a bit excessive for a paper, especially as over half are in the discussion. The discussion could do with cutting down by al least a half, if not more. The amount of text makes if difficult to sort out the final result and what they mean. There needs to be an explanation of each set of results, highlighting what each shows, then some refence to the literature followed by a final explanation relating all of the results together and explaining what they show. The fact that AA did not induce caspase activity, due to mopping up the H2O2 before it got to the cell, while the Mentha extracts were much less able to do this, hence the greater responsiveness of the cells. But the Authors then need to explain how this relates to the increase in Haem oxygenase with 250mM H2O2 and 80mm AA in Fig. 4a? Fig 4 shows H202 increase both pro-apoptotic BAx and anti-apoptotic Bcl Xl but, for example, Md1-3 decreases both? All of this needs to be commented on by the Authors and explained clearly against the existing literature. Because of the issue around what the letters mean on the graphs it is hard to confirm what si statically different. This is not helped by the Authors referring (e.g line 394) “”.. H2o2 increased the expression level …non-significantly” . Using a statistical test allows you to tell if something is increased significantly or not at all. The Authors need to be more careful in their statements, for example (line 356) surely the results show that activation of caspases was decreased, not the activity? The Authors need to reflect that the concentrations used are non -physiological and may be hard to generate, never mind maintain, in the human brain, even if the protective material could pass both the gut and the brain barriers.Reviewer 2 Report
This manuscript clarifies the neuroprotective effect of methane species against H2O2-induced oxidative stress and apoptosis in SH-SY5Y cells, and suggests that mint extract may lead to the therapeutic approach of AD.
Although this manuscript focuses mainly on the evaluation of antioxidant activity of mint extracts, it has already been clarified that mint extracts have ROS scavenging activity. Therefore, the research results of this manuscript are not novel.
If a neuroprotective activity of Mentha species is to be evaluated, an experimental system in which apoptosis is induced by adding Aβ40 and Aβ42 peptides to SH-SY5Y cells should be used.
In the Materials and Methods section, Fig1 and Fig2, the kits and manufacturers used for The BACE inhibition assay and the amyloid β (1-42) aggregation inhibition assay are not described. Therefore, it is unclear whether BACE inhibition activity and Aβ42-aggregation inhibition activity were measured by in vitro or cell free system.